# Impact of COVID-19 social distancing measures on future incidence of invasive pneumococcal disease in England and Wales: a mathematical modelling study

Yoon Hong Choi  ,[1] Elizabeth Miller[2]

[1]Statistics, Modelling and Economics Department, Public Health England, London, UK
[2]Department of Infectious Disease Epidemiology, London School of Hygiene & Tropical Medicine, London, UK

**Correspondence to**
Dr Yoon Hong Choi;
yoon.choi@phe.gov.uk

## ABSTRACT

**Objectives** In January 2020, the UK moved to a 1+1 schedule for the 13-valent pneumococcal conjugate vaccine (PCV13) with a single priming dose at 3-month and a 12-month booster. We modelled the impact on invasive pneumococcal disease (IPD) out to 2030/2031 of reductions in PCV13 coverage and population mixing associated with restrictions on non-essential healthcare visits and social distancing measures introduced in 2020/2021 to reduce SARS-CoV-2 transmission.

**Design** Using an existing model of pneumococcal transmission in England and Wales, we simulated the impact of a 40% reduction in coverage and a 40% reduction in mixing between and within age groups during two lockdowns in spring 2020 and autumn/winter 2020/2021. More and less extreme reductions in coverage and mixing were explored in a sensitivity analysis.

**Main outcome measures** Predicted annual numbers of IPD cases under different coverage and mixing reduction scenarios with uncertainty intervals (UIs) generated from minimum and maximum values of the model predictions using 500 parameter sets.

**Results** The model predicted that any increase in IPD cases resulting from a reduction in PCV13 coverage would be more than offset by a reduction in pneumococcal transmission due to social distancing measures and that overall reductions in IPD cases will persist for a few years after resumption of normal mixing. The net reduction in cumulative IPD cases over the five epidemiological years from July 2019 was predicted to be 13 494 (UI 12 211, 14 676) all ages. Similar results were obtained in the sensitivity analysis.

**Conclusion** COVID-19 lockdowns are predicted to have had a profound effect on pneumococcal transmission resulting in a reduction in pneumococcal carriage prevalence and IPD incidence for up to 5 years after the end of the lockdown period. Carriage studies will be informative in confirming the predicted impact of the lockdown measures after they have been lifted.

## STRENGTHS AND LIMITATIONS OF THIS STUDY

⇒ For this study, we used a previously published dynamic model of pneumococcal transmission that was fitted to past data on age-specific pneumococcal carriage and invasive pneumococcal disease (IPD) and was able to reproduce the observed herd immunity effects of the pneumococcal conjugate vaccination programme in England and Wales. Short-term model predictions of the effect of the first lockdown were validated against IPD data in England and Wales to June 2020.

⇒ Our conclusions were robust under the extreme scenario of assuming only a 10% reduction in mixing as a result of the two lockdowns and suspension of all 13-valent pneumococcal conjugate vaccine vaccination during the lockdown periods with no subsequent catch-up vaccination.

⇒ A limitation of our model is uncertainty over the future behaviour of non-vaccine serotypes, although this is unlikely to affect the overall conclusions.

⇒ The predicted effect of the reduction in social mixing on carriage of pneumococci could potentially reduce development of natural immunity as a result of carriage, but we were unable with the current model structure to investigate this.

## INTRODUCTION

Following the first identification of local transmission of SARS-CoV-2 in England on 28 February 2020,[1] the UK government instituted a series of control measures to reduce spread of the virus. On 18 March, schools along with restaurants and places of entertainment and leisure were closed, and on 23 March, all non-essential travel was banned with members of the population only permitted to leave their home for exercise once a day, or in order to obtain food or medication.[2] The public was advised to only use the National Health Service when essential,[3] and many general practices replaced face-to-face consultations with an online service.[4] This first national lockdown was eased in stages starting in May 2020 with children returning to schools in September 2020.[5] As COVID-19 cases started to increase again in September 2020, a variety

of locally based measures were introduced in areas of increasing incidence in an attempt to reduce transmission. Despite these local measures, cases continued to increase and a second national lockdown came into force in November 2020, which lasted for 6 months until May 2021.[5]

In the UK, paediatric vaccinations are given in general practice and there was initial concern that the measures taken to limit spread of SARS-CoV-2 would result in many children's vaccinations being postponed with a resulting increase in vaccine-preventable infections.[6] Indeed, early coverage data for measles/mumps/rubella vaccine for the first 3 weeks of the first lockdown in London suggested a drop of over 40% compared with the same period in 2019.[7] Of particular concern is the impact of a drop in coverage of the 13-valent pneumococcal conjugate vaccine (PCV13) for which the national 2+1 schedule (2, 4 and 12 months) was changed on 1 January 2020 to a 1+1 schedule at 3 and 12 months.[8] The rationale for the schedule change was that after a decade of high coverage with PCV13 much of the protection of young children against vaccine-type invasive pneumococcal disease (IPD) is through herd immunity generated by the 12-month booster dose and that dropping a priming dose would have little effect on this. The supporting evidence for the schedule change came from an immunogenicity study showing similar postbooster antibody responses to PCV13 serotypes with a 2+1 and 1+1 schedule[9] and a modelling study, which predicted little impact on IPD cases in children or adults if protection against vaccine-type carriage was similar postbooster with the two schedules.[10]

Using our previously described pneumococcal transmission model,[10] we investigated the impact on IPD if PCV13 coverage was substantially reduced during the lockdown periods in the UK. The model also allowed us to investigate the likely longer-term impact on pneumococcal transmission of social mixing restrictions put in place to limit the spread of the SARS-CoV-2 virus in England and Wales.

## MODEL AND ASSUMPTIONS

The same compartmental deterministic model and estimated parameter values as described by Choi et al[10] were used in this study. Briefly, the model simulates the transmission dynamics of carried pneumococci between and within different age groups and predicts carriage changes by serotype grouping, either vaccine type (VT) or non-vaccine type (NVT), under different vaccination scenarios. Carriage changes are then translated into IPD cases using case-carrier ratios (CCRs) that describe the invasive potential of different carried serotypes. The model has a susceptible–infected–susceptible (SIS) structure, as it is assumed that a carriage episode does not generate any serogroup-specific protection. The model has 26 parameters that describe the propensity for vaccine-induced replacement of VTs with NVTs in carriage, the CCRs in different age groups, and the effectiveness of PCV vaccination against

carriage of different vaccine serotypes (see online supplemental appendix for summary of model description and parameter estimation). The maximum likelihood values of these parameters and their uncertainty intervals (UIs) are estimated by fitting the model to IPD data up to the end of the epidemiological year from July 2015 to June 2016.[10] In implementing the effect of the 1+1 schedule, the base case in the model described by Choi et al[10] assumed that one priming dose at 3 months of age provides half the protection against VT carriage as two doses, while the effect of the 12-month booster remains unchanged. In the current analysis, the change to the 1+1 PCV13 schedule was implemented to reflect the actual start date of January 2020, not the assumed date of September 2018 as in the original model.[10]

We assumed that, during the first 2-month lockdown period starting from 23 March 2020 and also during the second 6-month lockdown starting from 5 November 2020, there was a 40% reduction in PCV13 coverage, as suggested by early declines in Measles, Mumps and Rubella (MMR) vaccine coverage in London during the first lockdown.[7] For the sensitivity analysis, we assumed two additional scenarios, one with no material decline in PCV13 coverage (0% decline) and a second with cessation of all PCV13 vaccination during the lockdown periods (100% decline). The 0% scenario was chosen to reflect a situation in which there was a rapid catch-up of missed PCV13 vaccinations after the first lockdown with no reduction during the second, while the 100% reduction scenario was explored as an extreme scenario. We pessimistically assumed there would be no subsequent catch-up vaccination for children under the 40% and 100% reduction scenarios, as final coverage data for the affected birth cohorts will not be available until 2021/2022.

To implement the impact of reduced social mixing during each of the lockdown periods, we used the same mixing matrix as before[10] based on the POLYMOD study[11] and an additional contact survey among infants under 1 year[12] but reduced the number of contacts between and within each age group by 40% as a baseline scenario and explored 10% and 74% reduction scenarios for the sensitivity analysis. The 74% reduction scenario was based on early data from a study of the social contacts of 1356 UK adults between 24 and 27 March[13] compared with the historical rates documented in the POLYMOD study.[11] However, this study[13] could not assess the impact on contacts made by children, particularly those of preschool age who are the main drivers of pneumococcal transmission.[10] As the social distancing measures in children were less strict than adults, the estimated reduction in mixing would be lower in children and so we took the 74% scenario as the upper extreme scenario for the contact pattern reduction.

The results are shown as the median of the model outputs under the different scenarios. The UI for each scenario represents the minimum and maximum values generated with 500 model parameters sets obtained from Choi et al.[10] The impact on IPD of the temporary changes

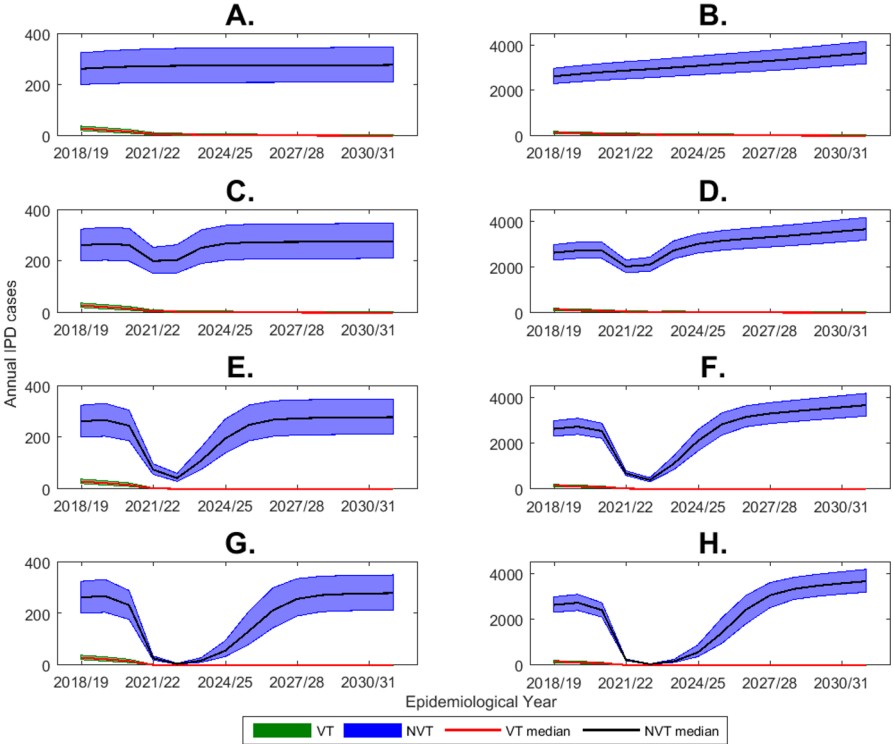

**Figure 1** Predicted annual number of IPD cases by VT and NVT and epidemiological year (July to June) from 2018/2019 to 2030/2031 with a 40% reduction in 13-valent pneumococcal conjugate vaccine coverage during the two lockdown periods starting from 23 March 2020 for 2 months and from 5 November 2020 for 6 months obtained from the long-term simulation model. (A) Under 5 year olds and (B) 65+ year olds without any reduction in contact rates. (C) Under 5 year olds and (D) 65+ year olds with 10% reduction in contact rates. (E) Under 5 year olds and (F) 65+ year olds with 40% reduction in contact rates. (G) Under 5 year olds and (H) 65+ year olds with 74% reduction in contact rates. Results show median and minimum and maximum of the range of model outputs generated by 500 model parameter sets.[10] IPD, invasive pneumococcal disease; NVT, non-vaccine type; VT, vaccine types.

in PCV13 coverage and social mixing is presented by VT, which includes PCV13 serotypes minus serotypes 1 and 3, and NVT; serotype 3 is included with the NVT group in the model on the grounds that it has shown an increase post-PCV13 similar to that seem for NVTs, while serotype 1 is excluded from the model due to secular changes which are independent of vaccine usage.[10]

### Patient and public involvement statement

Our study did not involve any direct participation by human subjects.

### RESULTS

Figure 1 shows the predicted number of IPD cases in England and Wales by epidemiological year out to 2030/2031 in under 5 year olds and in 65+ year olds by VT and NVT group with four different scenarios. The first scenario presented in figure 1A,B is without any change in PCV13 coverage and contact patterns; the increase in IPD cases out to 2030/2031 in 65+ year olds reflects the increasing mean age of individuals in this age group in the future together with the steep increase in IPD incidence with increasing age in the elderly population. The remaining three scenarios in figure 1C–H assume a 40% reduction in PCV13 coverage with respectively a 10% (C

and D), 40% (E and F) or 74% (G and H) reduction in contact rates during the two lockdowns. Even with only a 10% reduction in contact rates in figure 1C,D, the predicted impact of reduced social mixing outweighed any increase in IPD due to reduction in PCV13 coverage. The predicted beneficial effect of the lockdown on IPD cases was even more marked with higher reductions in social mixing (figure 1E–H). Under all the scenarios, the model predicted that the impact of the two lockdowns on IPD would persist for up to 5 years after their cessation.

The cumulative number of additional IPD cases predicted in England and Wales over the first 5 epidemiological years starting from July 2019 is shown in table 1 by age group and serotype grouping for a 40% reduction in contact rates and 40% reduction in PCV13 coverage during two lockdowns (base case scenario). Across all ages, the model predicted a net saving of 13 494 IPD cases (UI 12 211, 14 676) over this period compared with the predicted number of IPD cases without any lockdowns. The results over the first 5 years by age group and different scenarios for the reduction in coverage and social mixing are presented in online supplemental table S1. Without any reduction in contact rates under the worst case of no PCV13 vaccination during the two lockdown periods and no catch-up vaccination, the model predicted a

**Table 1** Cumulative difference in cases of invasive pneumococcal disease over 5 epidemiological years (July to June) starting from 2019/2020 by age group and serotype grouping (VT, NVT and overall) with a 40% reduction in 13-valent pneumococcal conjugate vaccine coverage, 40% reduction in contact rates during the two lockdown periods starting from 23 March for 2 months and from 5 November 2020 for 6 months

| | 0–1 years | 2–4 years | 5–14 years | 15–44 years | 45–64 years | 65+ years | Overall |
|---|---|---|---|---|---|---|---|
| VT | −13 (−26 to −6) | −2 (−4 to −1) | −8 (−15 to −3) | −82 (−164 to −36) | −94 (−186 to −41) | −156 (−312 to −69) | −354 (−707 to −156) |
| NVT | −456 (−564 to −334) | −167 (−206 to −116) | −268 (−343 to −249) | −1977 (−2195 to −1836) | −3282 (−3976 to −2844) | −6938 (−7956 to −6053) | −13123 (−14275 to −11893) |
| Overall | −471 (−578 to −347) | −169 (−208 to −118) | −275 (−349 to −256) | −2061 (−2291 to −1913) | −3367 (−4062 to −2931) | −7107 (−8135 to −6201) | −13494 (−14676 to −12211) |

Results show median and minimum and maximum of the range of model outputs generated by 500 model parameter sets.[10]
NVT, non-vaccine types; VT, vaccine types.

cumulative additional 105 (UI 49, 197) VT IPD cases, of which 14 (UI 6, 26) were in children under 2 years of age over the first 5 epidemiological years since 2019/2020. The overall predicted increase in IPD cases was smaller (73 (UI 33, 130)) due to a concomitant reduction in replacement with NVT IPD cases (figure 2). While the predicted overall increase is small, the perturbation in VT IPD cases as a result of the temporary interruption of vaccination is predicted to persist out to 2030/2031.

Amin-Chowdbury et al[14] reported that there were 2440 IPD cases in England and Wales between February and June 2019 and 1137 IPD cases during the same period in 2020 showing a 53% reduction in overall IPD cases in England and Wales. The model predicted that there would be 2350 (UI 2129, 2531) IPD cases in the same period in 2019.

Table 2 shows the model predictions for the reduction in IPD cases in February to June 2020 with a 40% reduction in PCV13 coverage during the two lockdown periods and under various reduced mixing scenarios. The scenario with a 40% reduction in mixing pattern most closely matched the observed reduction in IPD during February to June 2020.[14] Online supplemental table S2 shows the predicted number of IPD cases during February to June 2020 with different assumptions about reductions in PCV13 coverage and in social mixing. Only the scenarios with a 40% reduction in mixing produced the observed reductions seen in IPD incidence with little impact of varying the PCV13 coverage reductions.

## DISCUSSION

In line with many other countries, the UK adopted a lockdown strategy to minimise social contacts and thereby reduce the spread of the SARS-CoV-2. The restriction on non-essential travel and General Practice (GP) visits raised concerns that vaccine coverage in children might be substantially reduced,[6] echoing similar concerns raised at the global level.[15] With the recent change to a 1+1 PCV13 schedule in the UK, which is reliant on the booster dose to maintain herd immunity,[9 10] a substantial reduction in PCV13 coverage could compromise control of IPD. Our modelling study suggests that any potential impact of the lockdown measures on PCV13 coverage and herd immunity will be more than offset by a reduction in pneumococcal transmission due to the reduction in population mixing such that a net reduction in IPD cases should occur. Our model also predicts that the net reduction in IPD cases will be sustained for up to 5 years after mixing has returned to normal levels in the population. This reduction in IPD would be the result of an overall reduction in pneumococcal carriage prevalence after the two lockdown periods, unlike the IPD reduction achieved with PCV. Carriage studies in England before and after the introduction of PCVs have shown little impact on overall carriage prevalence with the reduction in carriage of vaccine serotypes offset by an increase in non-vaccine serotypes.[16] The overall reduction in IPD cases achieved by PCVs is the consequence of the

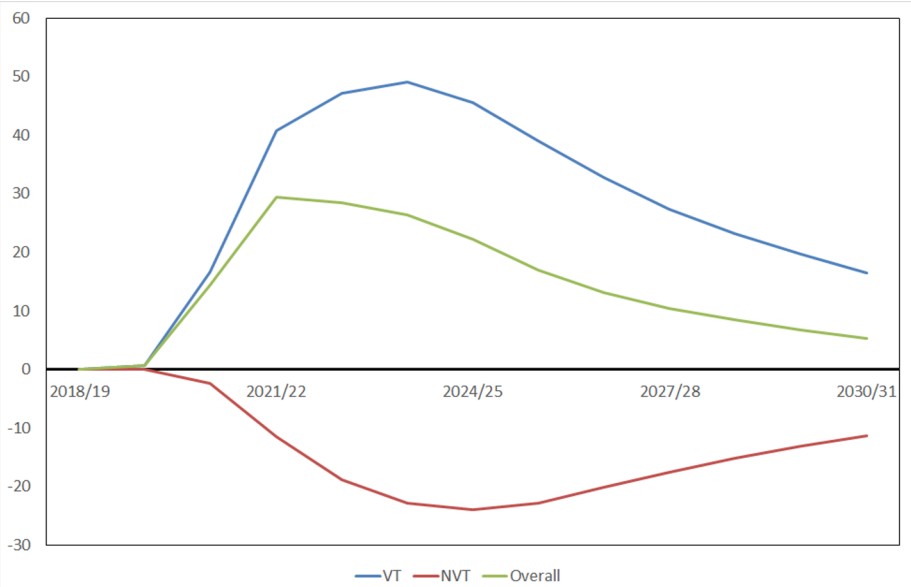

**Figure 2** Predicted median difference between the scenario without any change in 13-valent pneumococcal conjugate vaccine (PCV13) coverage and social mixing, and another scenario without any reduction in social mixing and no PCV13 vaccination during the two lockdowns in 2020/2021 in the annual number of IPD cases, by serotype grouping (VT, NVT and overall) in all ages in England and Wales. IPD, invasive pneumococcal disease; NVT, non-vaccine type; VT, vaccine types.

generally lower CCRs of the NVTs that replaced VTs in carriage.

IPD is a consequence, although rare, of carriage of *Streptococcus pneumoniae* in the nasopharynx. Acquisition of the pneumococcus is by person-to-person transmission and occurs via respiratory droplets with transmission enhanced in settings with close mixing such as day care centres and military camps.[17] Visits to the GP for mild upper respiratory tract disease have also been shown to be a risk factor for pneumococcal acquisition.[18] It is, therefore, to be expected that the measures taken to reduce spread of SARS-CoV-2 would also reduce pneumococcal transmission. The impact on IPD in England and Wales of the first UK lockdown that lasted for 2 months from 23 March 2020 was recently

reported.[14] Our model simulations of the impact of the first lockdown were similar and consistent with an overall reduction of 40% in contact patterns during this period.

Whether a reduction in coverage caused by the lockdowns would have a similar effect on other vaccine-preventable infections cannot be assessed with our model as it is specific to pneumococcal infection. However, the average number of weekly notifications of whooping cough and meningococcal septicaemia in England for 1 year from 23 March 2020 showed a 91.3% and 74.7% reduction, respectively, compared with the weekly average for the same period in 2019.[19] As with IPD, nasopharyngeal colonisation is a prerequisite for whooping cough and meningococcal disease. While the completeness of notifications by doctors may have suffered somewhat as a result of the COVID-19 epidemic, the magnitude of the decline in these two other infections is consistent with a substantial reduction in new colonisation episodes following the lockdown and suggests that, at least in the short term, this has outweighed any potential increase due to a reduction in coverage of pertussis or meningococcal vaccines.

Our model has limitations as previously described.[10] These include uncertainty over the future behaviour of non-vaccine serotypes and whether serotypes with different invasiveness potential will emerge, and also in the mixing patterns among the oldest age groups. However, given the magnitude of the predicted effect of the lockdowns, these uncertainties are unlikely to materially affect the results with respect to the effect of the lockdowns. It is also possible that the reduction in pneumococcal carriage predicted to occur as a result of the lockdowns may affect induction of natural immunity. Nasopharyngeal carriage can boost

**Table 2** Model outputs for the number of overall IPD cases between February and June 2020 and proportional reduction compared with the model predictions for the same period in 2019 (2350 (UI 2129, 2531) IPD cases) in England and Wales according to different reduction levels in contact patterns with 40% 13-valent pneumococcal conjugate vaccine coverage reduction rate during the first two lockdown starting from 23 March for 2 months in the UK

| Reduction in mixing | 2020 | Proportional reduction |
|---|---|---|
| 0% | 2374 (2146, 2554) | −0.9% (−1.3%, −0.6%) |
| 10% | 1954 (1766, 2102) | 16.9% (16.5%, 17.3%) |
| 40% | 1125 (1017, 1211) | 52.1% (51.8%, 52.4%) |
| 74% | 677 (612, 728) | 71.2% (71.1%, 71.3%) |

Results show median and minimum and maximum of the range of model outputs generated by 500 model parameter sets.[10]
IPD, invasive pneumococcal disease.

serotype-specific IgG antibody levels to the carried serotype and such naturally induced capsular antibodies have been shown to protect against carriage of that serotype.[20] If natural immunity against carriage was reduced as a result of decreased pneumococcal transmission this could offset the reduction in IPD predicted to persist after relaxation of the lockdown. Our model has an SIS structure so does not include a natural immunity compartment, so we were unable to explore the potential effect of such perturbations in carriage-induced immunity. We were also unable to incorporate the actual coverage reductions during the two lockdown periods as coverage data for completed 1+1 PCV13 courses is only available when a birth cohort reaches 2 years of age. Our baseline coverage reduction scenario may, therefore, have been extreme with catch-up vaccination rapidly offsetting any reduction during the two lockdown periods. However, as shown by the sensitivity analysis, the critical parameter in determining impact was reductions in mixing, with only a 10% reduction having a profound effect on IPD cases.

In conclusion, our pneumococcal model predicts that the effect of the COVID-19 lockdowns on pneumococcal carriage rates will result in a reduction in IPD cases over a 5-year period, thus obscuring the predicted small increase in VT IPD due to the adoption of a 1+1 PCV13 schedule.[10] Pneumococcal carriage studies conducted over the next few years could help elucidate the impact of the social distancing measures and of the adoption of the 1+1 PCV13 schedule on the future incidence of VT and NVT IPD.

**Contributors** YHC and EM conceived of the presented idea. YHC and EM developed the theory and YHC performed the computations. YHC and EM verified the analytical methods. YHC and EM discussed the results and contributed to the final manuscript.

**Funding** EM is funded by the National Institute for Health Research Health Protection Research Unit in Immunisation at the London School of Hygiene and Tropical Medicine in partnership with Public Health England (Grant Reference NIHR200929).

**Competing interests** None declared.

**Patient consent for publication** Not applicable.

**Provenance and peer review** Not commissioned; externally peer reviewed.

**Data availability statement** All data relevant to the study are included in the article or uploaded as supplementary information. All the data are available in our previous publication (https://journals.plos.org/plosmedicine/article?id=10.1371/journal.pmed.1002845).

**ORCID iD**
Yoon Hong Choi http://orcid.org/0000-0001-5561-4366

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
