## [Reviewer comments · BMJ Open]

ARTICLE DETAILS

TITLE (PROVISIONAL)	Impact of Covid-19 social distancing measures on future incidence of invasive pneumococcal disease in England and Wales – a mathematical modelling study
AUTHORS	Choi, Yoon Hong; Miller, Elizabeth

VERSION 1 – REVIEW

REVIEWER	Godelieve de Bree Amsterdam University Medical Centre, Department of Internal Medicine
REVIEW RETURNED	25-Feb-2021

GENERAL COMMENTS	General The authors describe the results of a modelling study that investigates the impact of lock down measures and social distancing on IPD cases, against the background of changes in vaccination regimen. In general this is a relevant subject and the data look solid. However, I have some comments especially regarding the choices made in factors that are used in the model. Furthermore, it would be good if the authors add the implications of their findings for vaccination strategies after the epidemic. For instance, would it be reason to vaccinate individuals who missed their vaccination (I can imagine based on their data this would make sense for the over 65 years age groups)? Major comments Model and assumptions The authors apply a few critical assumptions in the model that may request explanation as why some cut offs are used, the 2 most important ones are: Line 14 and 15 – 50% reduction in coverage: however in the introduction (see lines 16) the authors refer to a drop in coverage of 40%; Why was a 3 months and 6 lock down period selected given the protracted course of the pandemic and a much longer lock down period than 6 months. Reduction in contact rates by 10 and 50%, where was this based on, I would assume there are data available for “real world” reduction in contact rates among this age group. Results In figure 1 the effect of contact reduction is shown, it would help as a reference to have a figure with the model without reduced contacts (since in 2020 the vaccination regimen was changed) The authors show in figure 1 and table 1 that the effect of reduced
--

	vaccination coverage is overcome by distancing. But there remains a group of individuals that have not been vaccinated (I assume in the model they do not account for later catch up vaccination) and that may contract IPD in later years. How is this reflected in the model or data? This should be addressed in either the results or discussion section. Also, related to this comment: over the years there is an apparent increase in IPD in individuals over 65 years (see figure 1 and figure 1S), is this due to change in vaccination guidelines or a long term effect of the pandemic? Also for this interpretation it would be helpful to see the model data without reduction in contacts included. How long does the reduction in IPD last: in the discussion (line 15 p7) the authors state 5 years? This should also be added in the results section Minor comments Text line: ..in the model of the likely impact..: of should be omitted Discussion, lines 34-40 ..potential impact of social distancing policy on PCV13 coverage.. In the modelling section the authors separate social distancing and PCV coverage as separate factors. This sentence would need to be rephrased.
--	---

REVIEWER	Alessandra Løchen Imperial College London, Faculty of Medicine, School of Public Health Funded by an investigator-initiated grant from GlaxoSmithKline to my manager.
REVIEW RETURNED	21-Mar-2021

GENERAL COMMENTS	This study makes an important contribution to the ongoing research on the impact of COVID on other diseases, in this case pneumococcal disease. Overall, it is very clearly written and provides realistic model scenarios and interesting results. Minor comments: Introduction: Page 3, line 39: This sentence is quite awkward to read - consider revising to "The model also accounted for the likely impact of social mixing restrictions put in place to limit the spread of SARS-CoV-2" or something else that's easier to follow. Model and assumptions: In general, this section would benefit from spelling out some of the details of the model to clarify for the reader rather than being dependent on the other referenced paper - e.g. is this an SIS model? What CCRs were used? Are the age groups the same as the previous paper? Say what method was used for estimating maximum likelihood values. Would be good to have a table with the 26 parameter values mentioned in line 54 for transparency. It might be useful to say that the vaccination coverage statistics are not released until later which is why assumptions need to be made on the reduction in coverage. Page 4, line 37: define PCV13 serotypes I would also clarify over what time range the net IPD change is being
---

	looked at (and also include this in the Abstract), as you say five epidemiological years in the Results (Page 5, line 9) but then two years in the Discussion (Page 7, line 14). Results: I think it's worth including the extreme 100% reduction in vaccine coverage (Fig S1) in the main results rather than supplementary. Fig 1/S1/S2: Correct y-axis label to "Annual". Would it be possible to indicate the median on these plots as well? Page 5, line 11: fix quotation mark spacing for "lockdown" Table 1 is very interesting! The last sentences in this section "Without any reduction in contact rates...": it would be useful to have a table with the results that are being referred to here (at least in the supplementary) and some further comment on why there would be a concomitant reduction in NVT IPD cases in this worst case scenario. Discussion: Page 6, line 41: "many such notifications turn out not measles" did you mean "turn out not to be measles" General comments: Is there a need to continuously put lockdown in quotations, as it is a well-established term used by government and health officials?
--	---

VERSION 1 – AUTHOR RESPONSE

Reviewer: 1

Dr. Godelieve de Bree, Amsterdam University Medical Centre, Amsterdam Institute for Global Health and Development Comments to the Author:

General

The authors describe the results of a modelling study that investigates the impact of lock down measures and social distancing on IPD cases, against the background of changes in vaccination regimen. In general this is a relevant subject and the data look solid. However, I have some comments especially regarding the choices made in factors that are used in the model. Furthermore, it would be good if the authors add the implications of their findings for vaccination strategies after the epidemic. For instance, would it be reason to vaccinate individuals who missed their vaccination (I can imagine based on their data this would make sense for the over 65 years age groups)?

As answered in our response to the Associate Editor's comments above, simulation results indicate that all of these vaccine coverage reduction scenarios were offset by the reduction due to social distancing.

There is no PCV13 vaccination programme for the elderly UK. While there is a PPV23 vaccination programme this is a single dose to be given individuals become eligible at 65 years of age (ie given only once) so interruption of PPV23 administration in this age group would not materially affect IPD incidence in this age group.

Major comments

Model and assumptions

The authors apply a few critical assumptions in the model that may request explanation as why some cut offs are used, the 2 most important ones are:

Line 14 and 15 – 50% reduction in coverage: however in the introduction (see lines 16) the authors refer to a drop in coverage of 40%; Why was a 3 months and 6 lock down period selected given the protracted course of the pandemic and a much longer lock down period than 6 months.

As suggested by the reviewer, instead of 50% we implemented a 40% coverage reduction scenario as discussed above and also scenario in which there was no material reduction in coverage (0%) and a 100% reduction scenario in vaccine coverage (see response above on this point).

Reduction in contact rates by 10 and 50%, where was this based on, I would assume there are data available for "real world" reduction in contact rates among this age group.

We were aware of a study estimating the contact pattern change in the UK due to COVID_19 from Jarvis et al. [1]. However this study did not contain any children under 10-years of age who are the main drivers of the pneumococcal transmission dynamics. We used the data from this early study as an upper extreme scenario for the contact pattern reduction. For a sensitivity analysis, we considered 10% reduction as the least change in contact pattern and 40% reduction as a baseline scenario.

Results

In figure 1 the effect of contact reduction is shown, it would help as a reference to have a figure with the model without reduced contacts (since in 2020 the vaccination regimen was changed)

Thank you for the suggestion. We have now added figures for the scenario without any change in vaccine coverage and contact rates in Figure 1.

The authors show in figure 1 and table 1 that the effect of reduced vaccination coverage is overcome by distancing. But there remains a group of individuals that have not been vaccinated (I assume in the model they do not account for later catch up vaccination) and that may contract IPD in later years. How is this reflected in the model or data? This should be addressed in either the results or discussion section.

In order to illustrate the effect of having on its own a temporary interruption of vaccination with no catch-up (our 100% coverage reduction scenario) we have now included a new Figure 2 in which we show the difference in VT and NVT IPD in the population if there was 100% reduction in coverage during the two time periods corresponding to the lockdowns but without any reduction in social mixing compared with the scenario of no change in coverage and mixing. Unfortunately the model is not an IBM so does not allow us to track the cases in the actual birth cohorts who missed vaccination but overall the effect on the number of VT cases is relatively small though circulation of vaccine serotypes would persist for longer in the population (at least out to 2030/31) than without the temporary interruption of vaccination. We have now commented on this in the paper.

-

Also, related to this comment: over the years there is an apparent increase in IPD in individuals over 65 years (see figure 1 and figure 1S), is this due to change in vaccination guidelines or a long term effect of the pandemic? Also for this interpretation it would be helpful to see the model data without reduction in contacts included.

This is caused by an ageing population as the number of very elderly people in the 65+ age group who have the highest IPD attack rate is predicted to increase in the future. We have now included results from the scenario without any lockdown in Figure 1 A&B and hopefully explain more clearly the reason for the rising incidence in the 65+ year olds.

How long does the reduction in IPD last: in the discussion (line15 p7) the authors state 5 years? This should also be added in the results section

Simulation results indicated that the profound impact of lockdowns would continue over five years as presented in Fig 1. We added a sentence in Results as follows:

Our model also predicts that the net reduction in IPD cases will be sustained for up to five years after mixing has returned to normal levels in the population.

Minor comments

Text line: ..in the model of the likely impact..: of should be omitted

We omitted "of" as suggested.

Discussion, lines 34-40 ..potential impact of social distancing policy on PCV13 coverage.. In the modelling section the authors separate social distancing and PCV coverage as separate factors. This sentence would need to be rephrased.

Hopefully we have now clarified this sentence. The impact of the lockdown is two fold: 1. on coverage as non-essential GP visits stopped and 2. on mixing as a result of the social distancing measures. We rephrased as follows:

Our modelling study suggests that any potential impact of the lockdown measures on PCV13 coverage and herd immunity will be more than offset by a reduction in pneumococcal transmission due to the reduction in population mixing such that a net reduction in IPD cases should occur.

Reviewer: 2

Dr. Alessandra Løchen, Imperial College London Comments to the Author:

This study makes an important contribution to the ongoing research on the impact of COVID on other diseases, in this case pneumococcal disease. Overall, it is very clearly written and provides realistic model scenarios and interesting results.

Minor comments:

Introduction:

Page 3, line 39: This sentence is quite awkward to read - consider revising to "The model also accounted for the likely impact of social mixing restrictions put in place to limit the spread of SARS-CoV-2" or something else that's easier to follow.

We revised the sentence as follows:

The model also allowed us to investigate the likely longer-term impact on pneumococcal transmission of social mixing restrictions put in place to limit the spread of the SARS-CoV-2 virus in England and Wales.

Model and assumptions:

In general, this section would benefit from spelling out some of the details of the model to clarify for the reader rather than being dependent on the other referenced paper - e.g. is this an SIS model? What CCRs were used? Are the age groups the same as the previous paper? Say what method was used for estimating maximum likelihood values. Would be good to have a table with the 26 parameter values mentioned in line 54 for transparency.

The model is SIS which we have now indicated in the main text. We have also added a description of the model structure, age groups, data used for parameter estimation, fitting procedure and parameter estimation including CCR values in the Supplementary Appendix. This is an abbreviated version of the model description in our PloS Med 2019 paper which we hope can be understood by non-modellers. We have also included web links to the model design, parameter descriptions, CCR values and a table for the 26 parameter values for those who want to have easy access to the key model features in our original publication.

It might be useful to say that the vaccination coverage statistics are not released until later which is why assumptions need to be made on the reduction in coverage.

We added the following sentence in Discussion as suggested:

We were also unable to incorporate the actual coverage reductions during the two lockdown periods as coverage data for completed 1+1 PCV13 courses is only available when a birth cohort reaches two years of age.

Page 4, line 37: define PCV13 serotypes

I would also clarify over what time range the net IPD change is being looked at (and also include this in the Abstract), as you say five epidemiological years in the Results (Page 5, line 9) but then two years in the Discussion (Page 7, line 14).

We have now indicated in the abstract the time period over which our results are presented and that the main effects predicted to occur over the first five years after the lockdowns.

Results:

I think it's worth including the extreme 100% reduction in vaccine coverage (Fig S1) in the main results rather than supplementary.

As presented in Table S1, the extreme 100% reduction in vaccine coverage does not show any material difference to the 0% and 40% coverage reduction scenarios as they are dwarfed by the effects of the reduction in social mixing. We have however included a new table Fig 2 to illustrate the effect of stopping all PCV13 vaccination during the lockdown periods with no reduction in social mixing.

Fig 1/S1/S2: Correct y-axis label to "Annual". Would it be possible to indicate the median on these plots as well?

We corrected the y-axis label to "Annual" and added median lines in the figures as suggested.

Page 5, line 11: fix quotation mark spacing for "lockdown"

We removed all quotation marks as suggested. When we originally wrote the manuscript, the term "lockdown" was used colloquially but was not commonly used in the scientific literature but is now frequently used.

Table 1 is very interesting!

The last sentences in this section "Without any reduction in contact rates...": it would be useful to have a table with the results that are being referred to here (at least in the supplementary) and some further comment on why there would be a concomitant reduction in NVT IPD cases in this worst case scenario.

As stated above, we have introduced a new Figure 2 to show the predicted impact of temporarily stopping all PCV13 vaccination for the two time periods corresponding to when the lockdowns occurred but without any associated reduction in social mixing. While the overall increase in cases is relatively small, the perturbations of the reduction in coverage for the affected birth cohorts does persist out to 2030/31. We also present the net IPD changes with all combinations of assumptions in Table S1. It is a basic feature of the model that there is competition in the nasopharynx between carriage of VT and NVT serotypes so that as VT carriage increases, the NVT carriage will reduce and vice versa. We have now indicated this in the main text and described this more fully in the revised Supplementary Appendix.

Discussion:

Page 6, line 41: "many such notifications turn out not measles" did you mean "turn out not to be measles"

Yes, you are correct. We removed the mention of measles in the discussion as we felt it was a diversion. We also now have a whole year's of notification data for pertussis and meningococcal septicaemia which supports our model predictions on the impact of the reductions in social mixing on other infections acquired via the nasopharyngeal route.

General comments:

Is there a need to continuously put lockdown in quotations, as it is a well-established term used by government and health officials?

Yes agreed – see above.

VERSION 2 – REVIEW

REVIEWER	Bo Wang University of Leicester
REVIEW RETURNED	27-Jul-2021

GENERAL COMMENTS	The paper studies the impact of reduction in PCV13 coverage and population mixing during two lockdowns in spring 2020 and autumn/winter 2020/21. There are no new statistical methods and analyses involved in the study, and the mathematical model used is the same compartmental deterministic model as described by Choi et al. [10] and published in PLOS Med 2019, where the details of the model, the parameter estimation and fitting procedures are presented. The manuscript provides a good brief summary of the model structure, the parameters and the fitting procedures.
---

REVIEWER	James Odei Ohio State University, Division of Biostatistics
REVIEW RETURNED	03-Aug-2021

GENERAL COMMENTS	1. Page 5, Introduction: In line 22-23, please put a comma after the word “measures”2. Page 8, Results, Table 1: In Column 6, is the header 45-64Y or 45-65Y? Please check and correct this. I believe it should be 45-64Y to avoid an overlap with 65Y+ in Column 7.3. Page 9, Results: In line 31, please remove “. s” after the word “months”4. Page 20, Serotype groupings: In line 48-49, please remove the extra space before the word “and”
--

VERSION 2 – AUTHOR RESPONSE

Reviewer: 3
Dr. James Odei, Ohio State University
Comments to the Author:

The paper studies the impact of reduction in PCV13 coverage and population mixing during two lockdowns in spring 2020 and autumn/winter 2020/21. There are no new statistical methods and analyses involved in the study, and the mathematical model used is the same compartmental deterministic model as described by Choi et al. [10] and published in PLOS Med 2019, where the details of the model, the parameter estimation and fitting procedures are presented. The manuscript provides a good brief summary of the model structure, the parameters and the fitting procedures.

Author response:
We thank to Reviewer 4 for his kind comments.

Reviewer: 4
Dr. James Odei, Ohio State University
Comments to the Author:

1. Page 5, Introduction: In line 22-23, please put a comma after the word “measures”

Author response:

We put a comma after the word “measures” as Reviewer suggested.

2. Page 8, Results, Table 1: In Column 6, is the header 45-64Y or 45-65Y? Please check and correct this. I believe it should be 45-64Y to avoid an overlap with 65Y+ in Column 7.

Author response:

We corrected the header 45-65Y to 45-64Y as Reviewer suggested.

3. Page 9, Results: In line 31, please remove “. s” after the word “months”

Author response:

We removed “. s” after the word “months” as Reviewer suggested.

4. Page 20, Serotype groupings: In line 48-49, please remove the extra space before the word “and”

Author response:

We removed the extra space before the word “and” as Reviewer suggested.